# Harnessing the Power of LLMs: Evaluating Human-AI Text Co-Creation through the Lens of News Headline Generation

**Zijian Ding[1], Alison Smith-Renner[2], Wenjuan Zhang[2],**
**Joel R. Tetreault[2], Alejandro Jaimes[2]**
[1]University of Maryland, College Park, [2]Dataminr Inc.

## Abstract

To explore how humans can best leverage LLMs for writing and how interacting with these models affects feelings of ownership and trust in the writing process, we compared common human-AI interaction types (e.g., guiding system, selecting from system outputs, post-editing outputs) in the context of LLM-assisted news headline generation. While LLMs alone can generate satisfactory news headlines, on average, human control is needed to fix undesirable model outputs. Of the interaction methods, guiding and selecting model output added the most benefit with the lowest cost (in time and effort). Further, AI assistance did not harm participants' perception of control compared to freeform editing.

## 1 Introduction

Recent advancements in Large Language Models (LLMs), including the Generative Pre-trained Transformer (GPT) series (Brown et al., 2020), have shattered the previous ceiling of human-like text generation (Bommasani et al., 2021). This has led to a paradigm shift in NLP tasks, with task-agnostic LLMs surpassing the performance of other state-of-the-art task-specific models (Lee et al., 2022). However, LLM-backed systems are not without their flaws, often suffering from hallucinations, bias, or occasional generation of inappropriate content, such as toxic, discriminatory, or misleading information (Wang et al., 2022).

Human-AI text co-creation—or writing with AI assistance—allows some control over the generation process and the opportunity to overcome some LLM deficiencies. Co-creation approaches have shown tremendous promise in areas such as summarization (Goyal et al., 2022; Bhaskar et al., 2022) and creative writing (Moore et al., 2023; Cao, 2023; Yuan et al., 2022; Ding et al., 2023). Yet, the lack of *accountability* of AI (Shneiderman, 2022) shifts the burden of responsibility to humans when mistakes arise in the text co-creation process.

There are several methods of human-AI interaction for writing, which vary in terms of effort and control afforded by each (Cheng et al., 2022). For example, humans might simply select from alternative model outputs or have the flexibility to revise them (e.g., post-editing). Recent research by Dang et al. (2023) explored the dynamics of interaction between crowdworkers and Large Language Models (LLMs) within the scenario of composing short texts. Extending from their work, we evaluate human-AI interaction in the context of news headline creation, focusing on three of the interaction types identified by Cheng et al. (2022)—*guidance*, *selection*, and *post-editing*. Our study advances Cheng et al.'s Wizard-of-Oz framework by using an implemented AI system, providing more realistic insights into human-AI co-creation for news headline generation.

We explore three main aspects of co-creation with LLMs, through the lens of headline generation: which type of assistance is most effective for helping humans write high-quality headlines (RQ1)? Which type of assistance can reduce perceived effort for such task (RQ2)? And, how does interacting with these models affect trust and feelings of ownership of the produced headlines (RQ3)?

In a controlled experiment, 40 participants wrote news headlines–either manually or assisted by one of three variants of an LLM-powered headline writing system: (1) *selection*, (2) *guidance + selection*, and (3) *guidance + selection + post-editing*. To explore the balance between control and effort in tasks involving human-AI interaction (Figure 1), we examined the different interaction types in combination rather than separately. Next, 20 evaluators rated the headlines generated by five methods— manual, AI-only, and the three assisted conditions.

While LLMs alone generated the highest quality headlines on average, they were not perfect, and human control is needed to overcome errors. The simpler interactive condition *guidance + selec-*

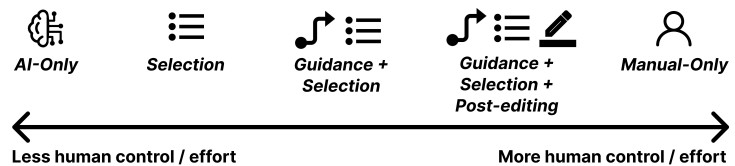

Figure 1: Human-AI interactions for text generation can fall within a range of no human control effort (AI-only) to full human control effort (manual methods) (Ding and Chan, 2023). Selecting from model outputs (*Selection*) alone provides less control (but also is easier) than when adding additional interactions in the form of guiding the model (*Guidance*) and *post-editing* the outputs.

*tion* resulted in the rapid creation of high-quality headlines compared to conditions involving further human intervention (i.e., post-editing or manual writing). All conditions yielded similar perceived trust and control in our task.

This work contributes: (1) a systematic evaluation of three variants of human-AI interaction for headline generation, (2) design guidelines for builders of these systems, and (3) a dataset[1] of 840 human-rated news headlines–written by humans, AI, or the combination, which could be used as an evaluation set for further leveraged with reinforcement learning, e.g., RLHF (Stiennon et al., [n. d.]), to adapt LLMs for news headline generation tasks.

## 2 Related Work

Our study draws upon a convergence of prior research on news headline generation and evaluation, and human-AI interactions for text summarization.

### 2.1 News Headline Generation and Evaluation

News headline generation, conventionally considered as a paradigm of text summarization tasks, has been extensively researched (Tan et al., 2017; Goyal et al., 2022). Advances in automation range from heuristic approaches like parse-and-trim (Dorr et al., 2003) to sophisticated machine learning algorithms like recurrent neural networks (Lopyrev, 2015), Universal Transformer (Gavrilov et al., 2019), reinforcement learning (Song et al., 2020; Xu et al., 2019), large-scale generation models trained with a distant supervision approach (Gu et al., 2020), and large language models (Zhang et al., 2023). Zhang et al. (2023) demonstrated that news summaries generated by freelance writers or Instruct GPT-3 Davinci received an equivalent level of preference from human annotators. In these studies, human involvement is for evaluation, not

[1]https://github.com/JsnDg/EMNLP23-LLM-headline.

creation. For human-AI co-creation in the context of journalism, a recent study explored the potential of harnessing the common sense reasoning capabilities inherent in LLMs to provide journalists with a diverse range of perspectives when reporting on press releases (Petridis et al., 2023).

### 2.2 Human-AI Interactions for Text Summarization

The taxonomy from Cheng et al. (2022) of human-AI interactions for text generation serves as the framework for our investigation. We focus on the first three types – *guiding model output*, *selecting or rating model output*, and *post-editing*, which offer varying degrees of control over the output and require different levels of human effort. Other identified types, such as interactive writing, are outside the scope of this study.

*Guiding model output* entails human intervention in the automatic text generation process, such as specifying crucial parts of input text to be included in the output (Gehrmann et al., 2019), or providing semantic prompts or style parameters (Osone et al., 2021; Chang et al., 2021; Strobelt et al., 2022). Our study involves human guidance in headline generation by specifying keyword prompts. *Selecting or rating model output* involves humans choosing from multiple model outputs (Kreutzer et al., 2018; Rosa et al., 2020) or rating them (Lam et al., 2018; Bohn and Ling, 2021). In our experiment, participants chose from three generated headlines. *Post-editing* refers to humans editing generated text (Chu and Komlodi, 2017; Huang et al., 2020), which could be accompanied by additional supports, such as suggestions for edits (Liu et al., 2011; Weisz et al., 2021). Our study investigates inline post-editing of selected headlines without any additional support.

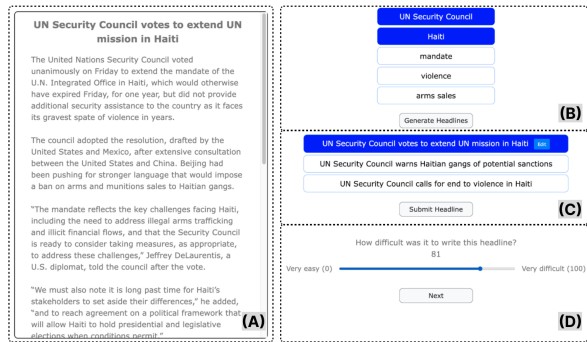

Figure 2: Interface for human-AI news headline co-creation for *guidance + selection + post-editing* condition: (A) news reading panel, (B) perspectives (keywords) selection panel (multiple keywords can be selected), (C) headline selection panel with post-editing capability, and (D) difficulty rating slider. Note: (B), (C) and (D) are hidden from the user until the requisite step is finished (e.g., the user does not see the difficulty rating slider until after finalizing the headline).

## 3 Assessing Human-AI Co-creation of News Headlines

We compared three variants of human-AI co-creation for news headlines against fully manual and fully automatic approaches. Specifically, we explored three research questions (RQs): **RQ1**: Which type of assistance is most effective for helping humans write high-quality headlines? **RQ2**: Which type of assistance can reduce perceived effort for such task? **RQ3**: How does interacting with these models affect trust and feelings of ownership of the produced headlines?

We chose news headline creation as it is a familiar task—many participants would already have some intuition about what constitutes a compelling (or not) news headline—of manageable complexity—both in task and evaluation, compared to free-form creative writing or long-form summarization (e.g., authoring paper abstracts).

### 3.1 Co-creation Methods, Implementation, and Prompts

Participants wrote headlines in one of four ways: manually or aided by AI using one of three common human-AI interaction types identified by previous research (Cheng et al., 2022): *guidance* of the LLM's headline generation using perspectives (keywords or facets), *selection* from three LLM-generated headlines, and *post-editing* the LLM-generated headlines.

Specifically, we developed a system for human-

AI co-creation of news headlines with four variants of human-AI interactions:

- *Selection*: The LLM generates three headlines for each news article (*generate headlines*), and the user selects the most appropriate one;

- *Guidance + Selection*: The LLM extracts several potential perspectives (keywords) from each news article (*extract perspectives*), the user chooses one or more perspectives to emphasize in the headline, the LLM then generates three headlines for each news article based on the selected perspectives (*generate headlines w/ perspectives*), and finally, the user selects the best one;

- *Guidance + Selection + Post-editing*: This is similar to *Guidance + Selection*, but the user can further edit the selected headline (post-editing);

- *Manual only*: The participant writes the news headline without any AI assistance.

Our goal was to create a continuum of human-AI interaction methods ranging from less to more control. The layering approach of selection, guidance, and post-editing was informed by prior literature (Cheng et al., 2022) and allowed us to assess the impact of each new layer by comparing adjacent conditions. Specifically, for these variants, we add additional functionality as opposed to comparing each interaction individually to allow for a more explicit comparison of conditions that afford less control (and require less effort)—e.g., *selection* only—opposed to more control (more effort)—e.g., *guidance + selection + post-editing* (Figure 1).

We used OpenAI's GPT-3.5 *text-davinci-002* model, the state-of-the-art LLM as of the study period (July 2022). Our study system directly interfaced with OpenAI's GPT-3.5 API, employing prompt programming to *extract perspectives* (keywords) and *generate headlines*—with and without supplied perspectives—and presented the results to the participants (see Figure 2).

To determine the optimal prompts for *extract perspectives*, *generate headlines*, and *generate headlines w/ perspectives* with the GPT-3.5 API, we conducted multiple configuration tests, ultimately selecting the zero-shot learning paradigm (with no examples of news headlines included) for extensive coverage of news topics. The GPT-3.5 API takes

prompts (provided in Appendix A) and generates the requested outputs.

To rigorously examine the LLM's reliability, we used a temperature setting of 1, ensuring maximum output variety. Additionally, we set a token count of 400, providing sufficient space for both input data—which included prompts and news articles—and output data, encompassing several keywords or news headlines.

## 3.2 Study Design

The study was structured into two phases: (1) human-AI co-creation of news headlines and (2) human evaluation of the generated headlines. The first phase utilized a between-subjects experimental design with four conditions, ranging from minimal to maximal human control: (1) *Selection*, (2) *Guidance + Selection*, (3) *Guidance + Selection + Post-editing*, and (4) *Manual only*. The second phase involved human evaluators who ranked the headlines generated in the first phase alongside the *original* article headlines and headlines generated by *GPT-only* without any human input.

### 3.2.1 Phase I: Human-AI Co-creation of Headlines

*Participants* We enlisted 40 participants from the freelancing platform, Upwork,[2] ensuring diversity in gender (28 females, 11 males, 1 non-binary). All the participants, who read news articles at least a few times a week if not daily, had experience in journalism, editing, or writing. Their educational backgrounds were diverse: 18 with Bachelor's Degrees, 11 with Master's Degrees, 10 with High School education, and 1 chose not to disclose.
*Procedure* Each participant was asked to create headlines for 20 articles,[3] published in June 2022 on Yahoo! News,[4] the most popular news website in the US during Q1 2022.[5] The articles were carefully chosen across four distinct themes: World Politics, COVID-19, Climate Change, and Science, with five articles each. The overall study procedure included an instruction round, a practice round, main study, and a post-task survey. Participants were randomly assigned to one of the four study conditions and compensated $20 for their 1-hour participation.

Participants first learned how to utilize the system with various types of AI assistance (*instruction round*) and wrote headlines for two practice articles (*practice round*).

During the *main study*, participants wrote headlines for the designated 20 task articles with assistance based on their assigned condition. The order of news articles was randomized for each participant. For each article, participants first read the article and clicked "Continue to Write Headline". Participants then either wrote the headline manually (*manual*) or with some assistance as described in Section 3.1. After completing each headline, participants rated the difficulty. Figure 2 demonstrates the system for the *Guidance + Selection + Post-editing* condition.

Finally, after writing headlines for all 20 articles, participants completed a *post-task survey*. See the Appendix for all study figures, including instructions, all conditions, and post-task survey.

### 3.2.2 Phase II: Human Evaluation of Headline Quality

*Participants* Another 20 evaluators were recruited through Upwork. We required that they were experienced in reviewing or editing news headlines to ensure high-quality evaluations.
*Procedure* In total there were 840 headlines requiring evaluation, including 800 headlines from Phase I (20 articles x 4 conditions x 10 participants per condition), 20 *Original* headlines, and 20 headlines generated by *GPT only*.

To ensure every headline was reviewed by at least two evaluators, we asked each evaluator to review 120 headlines–the six headlines for 20 articles—including 80 from Phase I, 20 *Original*, and 20 *GPT only*. We provided the evaluators with Excel forms containing instructions, news articles, and news headline ranking tables. For each article, the evaluators ranked the quality of six different headlines (the *Original*, *GPT Only*, and a headline generated by each of the four study conditions) using the Taste-Attractiveness-Clarity-Truth (TACT) test [6] from 1 (best) to 6 (worst) and shared their reasons:

- Taste: Is it in good taste? Is there anything possibly offensive in the headline? Can anything in the headline be taken the wrong way?

[2]https://www.upwork.com/
[3]The articles varied in length from 300 to 1000 words.
[4]https://news.yahoo.com/
[5]https://today.yougov.com/ratings/entertainment/popularity/news-websites/all

[6]http://www.columbia.edu/itc/journalism/isaacs/client_edit/Headlines.html

- Attractiveness: Is it attractive to the reader? Can it be improved so it is more engaging and interesting, without sacrificing accuracy?

- Clarity: Does it communicate the key points of the article? Is it clear and simple? Does it use the active voice and active verbs? Are there any odd words or double meanings that could confuse the reader?

- Truth: Is it accurate? Are the proper words or terms from the article used in the headline? Is the headline factually correct?

When two or more headlines for the same article were of similar quality, evaluators were able to assign the same ranking to them.[7] Each evaluator was compensated $20 each for this task, estimated to take around an hour.[8]

## 3.3 Measures

We measured headline quality (Section 3.2.2), perceived task difficult, headline creation time, and perceived control and trust of the system. For comparing efficiency between conditions, we care most about the headline creation time (from clicking "Continue to Write Headline" to "Submit Headline"). We further compute the article reading time (from starting to read the news article to clicking "Continue to Write Headline") and overall time (article reading time + headline creation time).

For user experience, we measured the task difficulty—after creating each headline (instance-level) and upon completing all headlines (overall), as well as perceived control and trust of the system. For instance-level task difficulty, participants answered "How difficult was it to create this headline for the news article with the system?" on a slider from 0 (very easy) to 100 (very difficult) after each headline writing task.

The remaining subjective measures were collected in the post-task survey: For overall task difficulty, participants agreed or disagreed with "It was difficult to create headlines for the news articles with the system." on a Likert scale from 1 (strongly disagree) to 5 (strongly agree) and answered a follow-up question, "why do you feel this way?". We operationalized perceived *control* and

trust of the system based on prior research: the question to gauge participants' perceived *control* was "I could influence the final headlines for the news articles" (Rader et al., 2018). The question to gauge perceived *trust* was "I trusted that the system would help me create high-quality headlines" (Cramer et al., 2008), via 5-point Likert rating scales (from strongly disagree to strongly agree).

## 3.4 Data Analysis

We employed the Kruskal-Wallis H test (Kruskal and Wallis, [n. d.]), the non-parametric version of a one-way ANOVA test for data with three or more conditions, to assess the quantitative results of headline quality rankings, task efficiency, and average and overall task difficulties, and perceived control and trust ratings. We then conducted a pairwise comparison of headline quality using the Mann-Whitney U test (Mann and Whitney, 1947), the non-parametric version of the Student t-test for two conditions. For headline quality, we compared the four conditions (*Manual*, *Selection*, *Guidance + Selection*, and *Guidance + Selection + Post-editing*) against the original headlines (*Original*) and the automatic method (*GPT only*). For task efficiency and user experience, we compared only within the four human study conditions. Finally, we analyzed participants' general comments and feedback on task difficulty.

## 4 Results

We compared headline quality (RQ1), task effort (RQ2) and perceived control and trust (RQ3) between the different headline writing conditions.

### 4.1 RQ1: Headline Quality

Rankings from *Phase II: Human Evaluation of Headline Quality* (Section 3.2.2) were used to assess headline quality across six conditions. Table 1 shows number of ranking data points and average rankings for each condition.[9]

As shown in Table 1 and 2, an approximate ranking of headline quality is as follows: *Original* ~ *GPT only* (best) > *Guidance + Selection* ~ *Guidance + Selection + Post-editing* > *Selection* ~ *Manual only*. The variance in quality rankings across the six conditions was significant, as confirmed by

---

[7]A small amount of headlines (1.3%, or 11/840) were found to be identical to another headline. These were due to participants selecting an identical GPT-generated headline without making further edits.

[8]Based on our pilot, evaluating one news article with six headlines took approximately three minutes.

[9]Rankings (from 1st to 6th place) for a total of 120 headlines (20 article x 6 conditions) were obtained (as shown in Table 1). However, ten were missed due to annotator error, resulting in 2390 total data points for analysis.

| Condition / Ranking | 1 | 2 | 3 | 4 | 5 | 6 | Mean | Std |
|---|---|---|---|---|---|---|---|---|
| *Manual only* | **56** | 60 | 56 | 56 | 71 | **98** | 3.8 | 1.8 |
| *Selection* | **56** | 53 | 65 | 67 | 83 | 75 | 3.7 | 1.7 |
| *Guidance + Selection* | 71 | 69 | 64 | 75 | 63 | 58 | 3.4 | 1.7 |
| *Guidance + Selection + Post-editing* | 70 | 71 | 74 | 60 | 65 | 58 | 3.4 | 1.7 |
| *Original* | **87** | 75 | 68 | 65 | 64 | **38** | 3.2 | 1.7 |
| *GPT only* | 70 | 73 | 89 | 79 | 51 | **37** | 3.2 | 1.6 |

Table 1: Counts of rankings from 1 (best) to 6 (worst) across six conditions, with mean rank and standard deviation. Lower values indicate superior rankings. Bold counts represent the highest number of 1st place rankings and the fewest 6th place rankings. The original headlines (*Original*) received the most 1st place rankings and the fewest 6th place rankings, while the *Manual* condition received the fewest 1st place and the most 6th place rankings.

| Condition 1 > Condition 2 | U-value | p-value |
|---|---|---|
| *Original > Manual only* | 61766.0 | <0.01 |
| *Original > Selection* | 63550.0 | <0.01 |
| *Original > Guidance + Selection* | 72364.5 | 0.028 |
| *Original > Guidance + Selection + Post-editing* | 72688.5 | 0.048 |
| *GPT only > Manual only* | 63168.5 | <0.01 |
| *GPT only > Selection* | 64751.5 | <0.01 |
| *Guidance + Selection > Manual only* | 89877.5 | <0.01 |
| *Guidance + Selection > Selection* | 88536.5 | <0.01 |
| *Guidance + Selection + Post-editing > Manual only* | 89972.5 | <0.01 |
| *Guidance + Selection + Post-editing > Selection* | 88734.0 | <0.01 |

Table 2: Pairwise comparison of headline quality rankings with significant difference (p<0.05, Mann-Whitney U).

Kruskal-Wallis H tests, with an $H-value = 50.8$ and a $p-value < 0.01$. Additional pairwise comparisons (detailed in Table 2) offered more granular insights into the discrepancies: headlines penned by participants in the *Manual* condition were deemed the lowest quality, whereas the original headlines and headlines generated by *GPT only* were deemed highest quality.

Within the AI-assisted conditions, human guidance, in the form of selecting keyword perspectives (under *Guidance + Selection* or *Guidance + Selection + Post-editing*), improved headline quality compared to direct *Selection* of AI-generated options. Interestingly, post-editing had no significant effect on the quality of the headlines: no noticeable difference in ranking was identified between the headlines generated by *Guidance + Selection* and *Guidance + Selection + Post-editing* conditions.

To delve deeper into group comparisons, we examined the count of "top" (ranked 1st) and "bottom" (ranked 6th) rankings (as shown in Table 1). As anticipated, the *Original* headlines of the news articles received the most "top" rankings (87) and the fewest "bottom" rankings (38). The *GPT only* condition had fewer "top" rankings than *Original* (70), but a comparable number of "bottom" rankings (37). The *Manual* and *Selection* conditions had the fewest "top" rankings (56), while *Manual* also had the most "bottom" rankings (98).

Evaluators comments gave additional insight into the reasons behind their rankings. For example, one annotator favored the concise headline generated by *GPT only* and critiqued the errors that undermined the *Manual* headline. For more details, see Appendix B.

## 4.2 RQ2: Task Effort

### 4.2.1 Headline Creation Time

We used task time as an indicator of task effort because it often reflects the workload someone experiences. Specifically, we compare *headline creation time*—the time spent to generate the news headline, exclusive of the time it took to read the news article. As expected, the *Selection* and *Guidance + Selection* conditions were markedly faster ($H-value = 13.0, p-value = 0.005$) than conditions that involved manual editing, such as *Manual* and *Guidance + Selection + Post-editing* (Figure 3). A detailed comparison of creation time, reading time, and total time is provided in Appendix C.

### 4.2.2 Perceived Task Difficulty

On the whole, participants felt the headline writing task was easy (e.g., average overall difficulty rating 1.9 out of 5). And, although *Guidance + Selection* had the lowest percieved instance-level difficulty (M=19.7 out of 100, SD=24.3) and *Manual* had the highest (M=30.9, SD=17.1), these differences were not statistically significant—in part due to high variance between participants and our small sam-

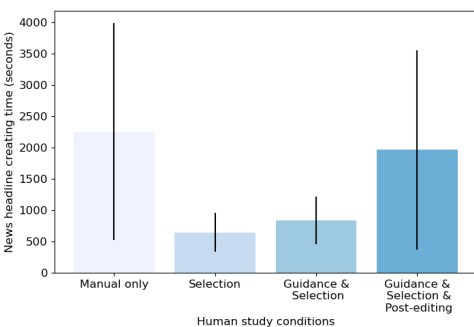

Figure 3: News headline creation time across four human conditions (standard deviation as error bars): *Selection* and *Guidance + Selection* are faster than the other two conditions which required more human editing (*Manual* and *Guidance + Selection + Post-editing*).

ple size. Additional results for instance-level and overall task difficulties are detailed in Appendix D.

### 4.3 RQ3: Perceived Trust and Control

We did not observe significant difference in perceived trust or control across the conditions (see Appendix E for more details). We had expected participants with more "control" over the final output (i.e., those writing manually or with option to *post-edit* the final output) to have more perceived control than the other conditions. Yet, even while most participants (9/10) in the *Guidance + Selection + Post-editing* condition did edit at least one headline during the task, with an average of 8.5 headlines edited ($median = 7.5, std = 6.0$), post-editing did not seem to enhance participants' sense of ownership over the generated headlines. We discuss possible reasons for this in Section 5.3.

## 5 Discussion

Our study explored how diverse forms of AI assistance for news headline writing influence headline quality (RQ1), task effort (RQ2), and perceived trust and control (RQ3). We interpret our results for each of the RQs and discuss further implications in the following.

### 5.1 RQ1: Which type of assistance helped humans write high-quality headlines?

Headlines produced by the *GPT only* condition were, on average, of the highest quality, whereas headlines written manually were of the lowest. And, all types of AI assistance explored in the study helped participants achieve higher quality as compared to when no assistance was available.

Therefore, system designers should consider leveraging these types of assistance to better support humans performing similar writing tasks.

While our findings could imply that completely automated methods are adequate for generating headlines, we urge system designers to critically evaluate when and where is appropriate to fully rely on AI assistance. Headlines by *GPT only* were not flawless, receiving the lowest ranking in some instances. Human intervention remains necessary for editing and final decisions, particularly in high-stakes tasks. Upon reflection, this study focused on overall headline quality in the evaluation task. Additional evaluation on the type, quantity, and severity of errors from headlines could reveal valuable insights. While multiple reasons could lead to a headline ranked as low quality, problematic hallucinations that are commonly seen with large language models (Bender et al., 2021; Ji et al., 2023) could be more harmful than omission of details.

Among the AI assistance, providing *Guidance* (using keywords) resulted in higher headline quality compared to *Selection* alone, yet did not outperform the *GPT only* condition.

Participants comments revealed that the quality of keywords given in the *Guidance* interaction played a crucial role in the final headline quality. When the keyword quality was poor, e.g., vague or not well aligned with key points of the article, the generated captions were lower quality as well. The design of the *Guidance* interaction may have limited its utility. One participant noted *"it would have been nice if I could go back and change the keywords and see how the headlines change"*, suggesting a more interactive experience could help them better understand how their guidance influences the model's output.

Headline evaluation is a subjective task, which can influence these outcomes. To mitigate this, we used the TACT evaluation framework as a standardized rubric, which was shared with both headline generators and evaluators. Importantly, human control over the final headline is a promising direction given the subjective nature (i.e., simply relying on the system to pick the *best* headline is problematic if what is *best* differs by the audience and their desired headline qualities.

| | Type | Before Revision | After Revision |
|---|---|---|---|
| 1) | Hedging | Is your lawn **ruining the environment**? | Is your lawn **contributing to climate change**? |
| 2) | Hedging | Soldiers given second change to get vaccinated and avoid **expulsion** | Soldiers given second chance to get vaccinated and avoid **penalties** |
| 3) | Catering | Mapping the Seafloor: New technologies crucial for **completion** | Mapping the Seafloor: New technologies crucial for **climate change** |
| 4) | Clarifying | Forest Service employees made several mistakes when planning a controlled burn in New Mexico, resulting in the largest fire in **the state's** history | Forest Service mistakes when planning a controlled burn in New Mexico, resulting in the largest fire in **New Mexico history** |

Table 3: Examples of humans' post-editing on AI-generated headlines.

## 5.2 RQ2: Which type of assistance reduced perceived effort?

In terms of headline creating time, interactions such as guiding the model output and selecting generated headlines were quicker than post-editing, as expected. This means when efficiency is key to a system's performance, these type of interactions would be most effective, especially when the model outputs are generally satisfactory. Post-editing would likely be more critical if the quality of the model's output was suboptimal, or when humans need the ability to fully control the outcome.

Interestingly, in perceived task difficulty did not differ across all conditions. This is likely because most participants were experienced in headline writing. While their experience level varied greatly, from working for school newspaper to being in journalism for more than 40 years, the hands-on experience made the task in our study easy for them. Future research direction might evaluate if such AI assistance can lower the expertise requirements for headline writing task, enabling people with less experience to write good quality headlines.

## 5.3 RQ3: How does interacting with these models affect feelings of trust and ownership of the produced headlines?

We did not observe any significant difference in perceived trust or control. While surprising, we expect this might be from a flaw in our operationalization of perceived control; the statement *"I could influence the final headlines for the news articles"* does not specify the level or type of control one had over the system, and the participants could "influence the final headlines" to some extent no matter which condition they experienced. We recommend future research investigate more robust methods for measuring perceived control or sense of ownership.

Another reason for the lack of significant difference in this (and other subjective measures) is that each participant only experienced one condition and had no way to compare with other conditions. A within-subject study would be more effective for comparing perceived difficulty, control, and trust across multiple experiences. However, we chose a between-subject design to avoid the fatigue of having to complete similar tasks with multiple interfaces and the confusion from switching from one experience to another.

Finally, participants may have felt they did not need to exert much control for the system to perform well and, therefore, did not think about the level of control critically. Counter to what the results showed, we had expected post-editing to improve the sense of control. Analysis of their post-editing process (Table 3) revealed that changes made by participants during post-editing were typically minor, falling into three categories:

1. "Hedging": Participants generally avoided sensationalism or "click-bait" language, favoring more measured phrasing. For instance, "ruining the environment" was altered to "contributing to climate change", and "expulsion" was revised to "penalties".

2. "Catering": Participants frequently tweaked the headlines to make them more relevant to the target audience, for instance, changing "completion" to "climate change".

3. "Clarifying": Some edits involved minor text adjustments, such as replacing "the state's" with "New Mexico".

Participants may have associated smaller degree of changes with smaller level of control to the system output. However, we highlight that these revision types could be supported natively by future systems.

## 5.4 Control, Transparency, and Guardrails in Co-Creation

The interaction options we introduced extend beyond mere control, instead providing both

guardrails and a way to expose what is possible with LLMs. First, both the *selection* and *guidance* interactions provide a set of guardrails toward better output opposed to freeform prompting of the LLM. Next, as participants can guide the model via theme or keyword inputs, they are also afforded a level of *transparency* to comprehend the influence they may exert on LLMs to yield headlines with varying focuses. These approaches which combine control and transparency seem critical to successful co-creation settings (Liao and Vaughan, 2023).

We did not evaluate a setting where users prompted the LLM themselves (this is a suggestion for future work); instead, our settings—which incorporated optimized prompts (Appendix A) —are meant to provide a more *directed* scenario, without requiring the user to come up with possible keywords from scratch. This scenario progressively exploded users to how the model works and how they can control it. Future iterations of the *guidance* interaction could benefit from additionally offering users a choice between system-suggested and user-defined keywords to flexibly steer the model.

# 6 Conclusion

We explore human-AI text co-creation through the context of news headline generation: how can humans best harness the power of LLMs in writing tasks, in terms of both quality and efficiency? And, how does interacting with these models affect trust and feelings of ownership of the produced text? 40 participants with editing backgrounds wrote news headlines—either manually or assisted by a variant of our LLM-powered headline generation system—utilizing some combination of three common interaction types (guiding model outputs, selecting from model outputs, and post-editing). Following this, 20 expert evaluators rated the headlines generated by five methods (human-only, AI-only, and the three assisted conditions). While the LLM on its own can generate satisfactory news headlines on average, human intervention is necessary to correct undesirable model outputs. Among the interaction methods, guiding model output provided the most benefit with the least cost (in terms of time and effort). Furthermore, AI assistance did not diminish participants' perception of control compared to freeform editing. Finally, while we focus on a simpler, low-stakes text generation scenario, this work lays the foundation for future research in more complex areas of human-AI co-creation.

# Limitations

This study, while comprehensive, acknowledges several limitations that outline areas for potential future research. One of the constraints pertains to the evaluation of different models. Specifically, the research primarily focused on types of human-AI interaction, utilizing GPT-3.5 due to its state-of-the-art performance during the research phase. The choice was made to prevent the complication of the study design by introducing performance as an additional variable, which could have overshadowed the main objective of analyzing three interaction variants. However, understanding the impact of varying models and their performances remains an essential prospect for subsequent studies and system design.

Moreover, in terms of evaluation measures, the study employed the Taste-Attractiveness-Clarity-Truth (TACT) framework to assess headline quality. Despite its standardization, this framework may not fully encapsulate all the nuances of a model's performance, such as verbosity, indicating a need for future work to refine these evaluation standards.

Additionally, the research's scope was limited in terms of the number and types of articles, as it prioritized the examination of interaction differences over article diversity. This limitation underscores the importance of future studies exploring a broader array of articles and involving more participants for an extended period. Similarly, the study did not evaluate the efficiency of the model in generating diverse perspectives, maintaining consistency in article selection based on length and domain instead. The assessment of content generation difficulty is identified as a crucial element for more in-depth research.

Concerning participant expertise, the study conducted was a concise, one-hour session centered on human-AI collaboration in news headline creation, taking into account participant expertise without enforcing strict control. This approach points to the necessity for more extensive research, particularly in more complex and specialized domains.

Finally, regarding study design and comparison, the research adopted a between-subjects approach, preventing participants from making direct comparisons of the different human-AI interaction methods. As a result, certain participants might not have fully grasped how AI could potentially enhance the task, as their experience was confined to their specific condition. This highlights an oppor-

tunity for future studies to possibly implement a within-subjects design for a more holistic participant experience.

## Ethics Statement

Although the performance of headlines generated by *GPT only* were comparable to the original headlines and those headlines did not demonstrate evident biases or ethical issues, there is an open question of to what extent AI should be relied upon. Shneiderman proposes that "we should reject the idea that autonomous machines can exceed or replace any meaningful notion of human intelligence, creativity, and responsibility" (Shneiderman, 2022). We echo that serious consideration is needed before AI is relied upon for creating text, even for low stakes tasks. It is important for humans to use their expertise and judgment when working on content generation with AI and ensure that the content produced is fair, ethical, and aligned with societal values.

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

## APPENDIX

## A  Prompts for tasks

To determine the optimal prompts for *extract perspectives*, *generate headlines*, and *generate headlines w/ perspectives* with the GPT-3.5 API, we conducted multiple configuration tests, ultimately selecting the zero-shot learning paradigm (with no examples of news headlines included) for extensive coverage of news topics.

Here are the finalized prompts for the corresponding tasks:

- *extract perspectives*: "Please identify five keywords and key phrases (nouns) for the following news article.\nNews article:\n + [news article] + \nFive keywords and key phrases (nouns), separated with comma:\n"

- *generate headlines*: "Please come up with three high-quality (attractive, clear, accurate and inoffensive) headlines for the following news article.\nNews article:\n\n + [news article] + \n\nThree high-quality headlines:\n"

- *generate headlines w/ perspectives*: "Please come up with three high-quality (attractive, clear, accurate and inoffensive) headlines for the following news article including keyword(s): [keyword(s) selected by participants].\nNews article:\n\n + [news article] + \n\nThree high-quality headlines:\n"

## B  Annotator Rationale Case Study

Table 4 illustrates the rankings assigned by an annotator to six different headlines, each generated under a unique condition, for a single article. Accompanying comments provide valuable insights into the annotator's ranking rationale.

## C  Task Time

Table 5 presents the time spent on different parts of the task—reading the article, creating the headline, and the total time for both reading and creating the headline—between conditions.

## D  Perceived Task Difficulty

Table 6 presents the self-reported instance-level and overall task difficulty ratings between conditions.

## E  Perceived Control and Trust

Table 7 presents the self-reported trust in the system (trust) and feelings of ownership over the final headline (control).

## F  Qualitative feedback on AI-assisted headline generation

After the headline generation task, participants were asked for open-ended responses as a supplement to their perceived task difficulty and were invited to share "any other comments you want to tell us". These responses illuminated perceived advantages of AI assistants, for example in facilitating brainstorming: "Overall, I'd say this system could definitely be helpful as long as the keyword list lines up with the main keywords in the article. Additionally, it would always be a helpful brainstorming tool for headlines." (P40, *Guidance + Selection + Post-editing*)

Some participants proposed specific considerations for the design of future AI-assisted systems: 1) **inclusion of main points in AI-generated headlines** "Some of the main points were left out of certain headlines, but otherwise, this proves to be a useful tool for journalists. The main points were combined into an easy-to-use database for selection." (P23, *Guidance + Selection*); 2) **high efficiency but limited creativity and reasoning** "I could see how this would be beneficial in situations where time is limited, but it does not replace the creativity and reasoning a human has. Some seemed spot on and others seemed chaotic. But with a few tweaks, it was easy to create a headline. Some of the keywords suggested seemed totally unnecessary. The keywords were the weakest part of the system in my opinion." (P39, *Guidance + Selection + Post-editing*); 3) **guidance on AI model and selection of AI outputs as a non-linear process** "I do wish I could have gone back to choose different perspectives and see alternate variations than just the 3 original headlines. If those first 3 options don't really cover the central theme of the article, it would be nice to go back and choose alternate perspectives to see if it populates a more precise headline. Overall though, this is a great tool to save time and many headaches. I really enjoyed this experience and would love to participate again in future exercises." (P28, *Guidance + Selection*)

| Type | Title | Rank | Comment |
|---|---|---|---|
| *Manual only* | National Army Guard Demands Soldiers to be Vaccinated against Covid-19 | 6 | The title of the Army is wrong and it's also not the subject of the article. |
| *Selection* | Army National Guard facing recruiting crisis amid vaccination deadline | 3 | This title does not include COVID which is a prominent subject and good for SEO considering the article is about it |
| *Guidance + Selection* | Army National Guard misses Covid vaccine deadline, faces recruiting crisis | 2 | This title is good as well. The topic of recruiting crisis should be out before the COVID vaccine deadline |
| *Guidance + Selection + Post-editing* | Army National Guard Members Refuse Vaccine, Face Potential Expulsion | 5 | This is not the main subject of the article. Just a highlight in the article |
| *Original* | Deadline passes and 1 in 10 Army National Guard soldiers still unvaccinated for Covid. Will they be expelled? | 4 | This title sums up the article but it needs to be more concise. The sentence is too long. |
| *GPT only* | Army National Guard Faces Recruiting Crisis Amid Covid Vaccine Deadline | 1 | Clear title - Contains subject and issue that has arised. |

Table 4: Representative evaluation task, showing one (of 20) annotators evaluation of one (of 20) articles. Headlines are ranked from best (1) to worst (6).

| Condition / Time Duration (s) | Reading | | Creating | | Total | |
|---|---|---|---|---|---|---|
| | mean | std | mean | std | mean | std |
| *Manual only* | 1711 | 953 | 2248 | 1733 | 3959 | 1887 |
| *Selection* | 2723 | 1503 | **644** | **313** | 3367 | 1417 |
| *Guidance + Selection* | 1998 | 1581 | **838** | **379** | 2835 | 1634 |
| *Guidance + Selection + Post-editing* | 1362 | 885 | 1961 | 1588 | 3324 | 1700 |
| | *H-value* | *p-value* | *H-value* | *p-value* | *H-value* | *p-value* |
| | 4.2 | 0.24 | **13.0** | **0.005** | 1.9 | 0.59 |

Table 5: Duration (in seconds) of reading time (the time spent on reading the news article), creating time (the time spent on generating the news headline), and total time. The differences were tested with Kruskal-Wallis H test.

| Condition / Difficulty | Instance-level | | Overall | |
|---|---|---|---|---|
| | mean | std | mean | std |
| *Manual only* | 30.9 | 17.1 | 1.9 | 1.0 |
| *Selection* | 25.0 | 19.4 | 1.9 | 0.7 |
| *Guidance + Selection* | 19.7 | 24.3 | 1.7 | 0.5 |
| *Guidance + Selection + Post-editing* | 20.3 | 25.1 | 1.9 | 0.7 |
| | *H-value* | *p-value* | *H-value* | *p-value* |
| | 4.0 | 0.26 | 0.4 | 0.94 |

Table 6: Participants self-reported instance-level task difficulty rating (average across all instances, from 0 to 100) and overall task difficulty rating (Likert-scale, from 1 to 5) for four conditions. In both cases, higher scores means more difficult.

| Statements | Manual only | Selection | Guidance Selection | Guidance Selection Post-editing | H-value | p-value |
|---|---|---|---|---|---|---|
| [Control] I could influence the final headlines for the news articles. | 3.8 (0.7) | 3.4 (1.3) | 3.9 (0.7) | 3.9 (1.2) | 1.6 | 0.66 |
| [Trust] I trusted that the system would help me create high-quality headlines. | 3.9 (1.1) | 3.6 (0.7) | 3.6 (0.5) | 3.7 (0.8) | 0.8 | 0.85 |

Table 7: Means, standard deviation (in parenthesis) and Kruskal-Wallis H test results of users' level of agreement to statements of participants' control and trust on a Likert scale of 1 (Strongly Disagree) to 5 (Strongly Agree) for Condition 0, 1, 2 and 3. No significant difference was observed for control or trust among the four study conditions.

## G   Interfaces for human study

The screenshots of the interfaces for the human study were attached below. All the practice rounds and human studies shared the same interfaces and interactions. We took the first round of practice as an example to demonstrate the interfaces for four AI-assisted conditions: *Manual only*, *Selection*, *Guidance + Selection*, and *Guidance + Selection + Post-editing*.

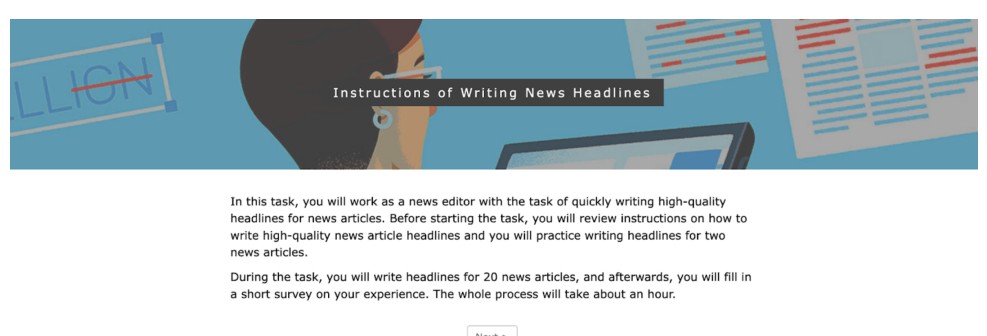

Figure G1: General task instruction for participants.

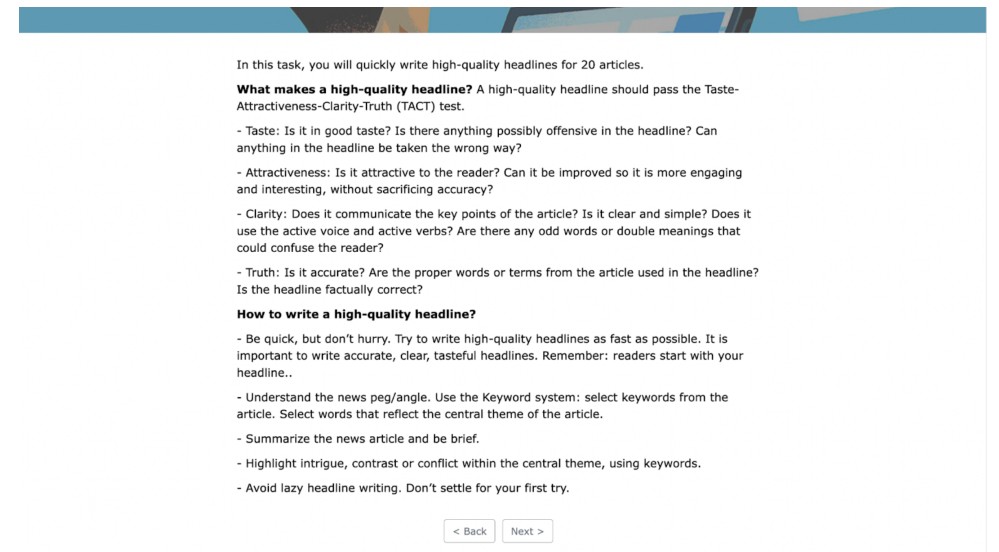

Figure G2: Instructions on how to generate high quality news headlines for participants.

Please read the news article below carefully.

The United Nations Security Council voted unanimously on Friday to extend the mandate of the U.N. Integrated Office in Haiti, which would otherwise have expired Friday, for one year, but did not provide additional security assistance to the country as it faces its gravest spate of violence in years.

The council adopted the resolution, drafted by the United States and Mexico, after extensive consultation between the United States and China. Beijing had been pushing for stronger language that would impose a ban on arms and munitions sales to Haitian gangs.

"The mandate reflects the key challenges facing Haiti, including the need to address illegal arms trafficking and illicit financial flows, and that the Security Council is ready to consider taking measures, as appropriate, to address these challenges," Jeffrey DeLaurentis, a U.S. diplomat, told the council after the vote.

"We must also note it is long past time for Haiti's stakeholders to set aside their differences," he added, "and to reach agreement on a political framework that will allow Haiti to hold presidential and legislative elections when conditions permit."

A representative of Mexico, Juan Ramón de la Fuente Ramirez, told the Security Council that the resolution "calls for a cessation of violence."

The U.N. political mission in Haiti was established in 2019 to promote stability in Port-au-Prince.

But the ongoing security crisis in Haiti, which has been devastated by gang violence and political paralysis since the assassination of its president last year, led some to call for a more robust security package to accompany the renewal.

Continue to Write Headline

Figure G3: Interface for news article reading (same for all conditions).

Please write a headline for the news article.

The United Nations Security Council voted unanimously on Friday to extend the mandate of the U.N. Integrated Office in Haiti, which would otherwise have expired Friday, for one year, but did not provide additional security assistance to the country as it faces its gravest spate of violence in years.

The council adopted the resolution, drafted by the United States and Mexico, after extensive consultation between the United States and China. Beijing had been pushing for stronger language that would impose a ban on arms and munitions sales to Haitian gangs.

"The mandate reflects the key challenges facing Haiti, including the need to address illegal arms trafficking and illicit financial flows, and that the Security Council is ready to consider taking measures, as appropriate, to address these challenges," Jeffrey DeLaurentis, a U.S. diplomat, told the council after the vote.

"We must also note it is long past time for Haiti's stakeholders to set aside their differences," he added, "and to reach agreement on a political framework that will allow Haiti to hold presidential and legislative elections when conditions permit."

A representative of Mexico, Juan Ramón de la Fuente

Please write a headline.

Submit Headline

Figure G4: Interface for *Manual only* condition of news headline generation.

**UN Security Council calls for cessation of violence in Haiti**

The United Nations Security Council voted unanimously on Friday to extend the mandate of the U.N. Integrated Office in Haiti, which would otherwise have expired Friday, for one year, but did not provide additional security assistance to the country as it faces its gravest spate of violence in years.

The council adopted the resolution, drafted by the United States and Mexico, after extensive consultation between the United States and China. Beijing had been pushing for stronger language that would impose a ban on arms and munitions sales to Haitian gangs.

"The mandate reflects the key challenges facing Haiti, including the need to address illegal arms trafficking and illicit financial flows, and that the Security Council is ready to consider taking measures, as appropriate, to address these challenges," Jeffrey DeLaurentis, a U.S. diplomat, told the council after the vote.

"We must also note it is long past time for Haiti's stakeholders to set aside their differences," he added, "and to reach agreement on a political framework that will allow Haiti to hold presidential and legislative elections when conditions permit."

The system has generated the following headline alternatives for the news article.

Please select the best headline and click "Submit Headline".

UN Security Council unanimously extends mandate for UN Integrated Office in Haiti

UN Security Council addresses illegal arms trafficking and illicit financial flows in Haiti

UN Security Council calls for cessation of violence in Haiti

Submit Headline

How difficult was it to write this headline?
70
Very easy (0) ———————————— Very difficult (100)

Next

Figure G5: Interface for *Selection* condition of news headline generation.

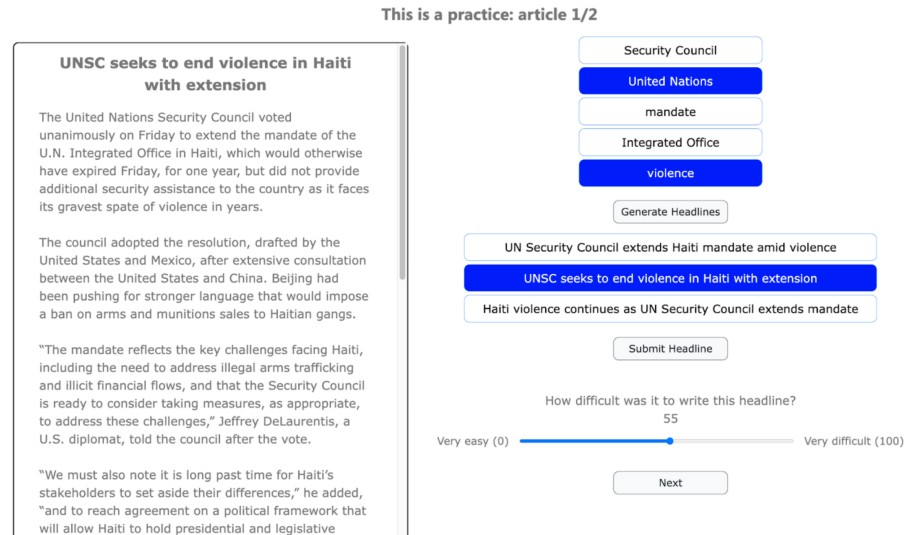

Figure G6: Interface for *Guidance + Selection* condition of news headline generation.

This is a practice: article 1/2

**UN Security Council votes to extend UN mission in Haiti**

The United Nations Security Council voted unanimously on Friday to extend the mandate of the U.N. Integrated Office in Haiti, which would otherwise have expired Friday, for one year, but did not provide additional security assistance to the country as it faces its gravest spate of violence in years.

The council adopted the resolution, drafted by the United States and Mexico, after extensive consultation between the United States and China. Beijing had been pushing for stronger language that would impose a ban on arms and munitions sales to Haitian gangs.

"The mandate reflects the key challenges facing Haiti, including the need to address illegal arms trafficking and illicit financial flows, and that the Security Council is ready to consider taking measures, as appropriate, to address these challenges," Jeffrey DeLaurentis, a U.S. diplomat, told the council after the vote.

"We must also note it is long past time for Haiti's stakeholders to set aside their differences," he added, "and to reach agreement on a political framework that will allow Haiti to hold presidential and legislative elections when conditions permit."

UN Security Council
Haiti
mandate
violence
arms sales

Generate Headlines

UN Security Council votes to extend UN mission in Haiti   Edit
UN Security Council warns Haitian gangs of potential sanctions
UN Security Council calls for end to violence in Haiti

Submit Headline

How difficult was it to write this headline?
81
Very easy (0)                              Very difficult (100)

Next

Figure G7: Interface for *Guidance + Selection + Post-editing* condition of news headline generation.

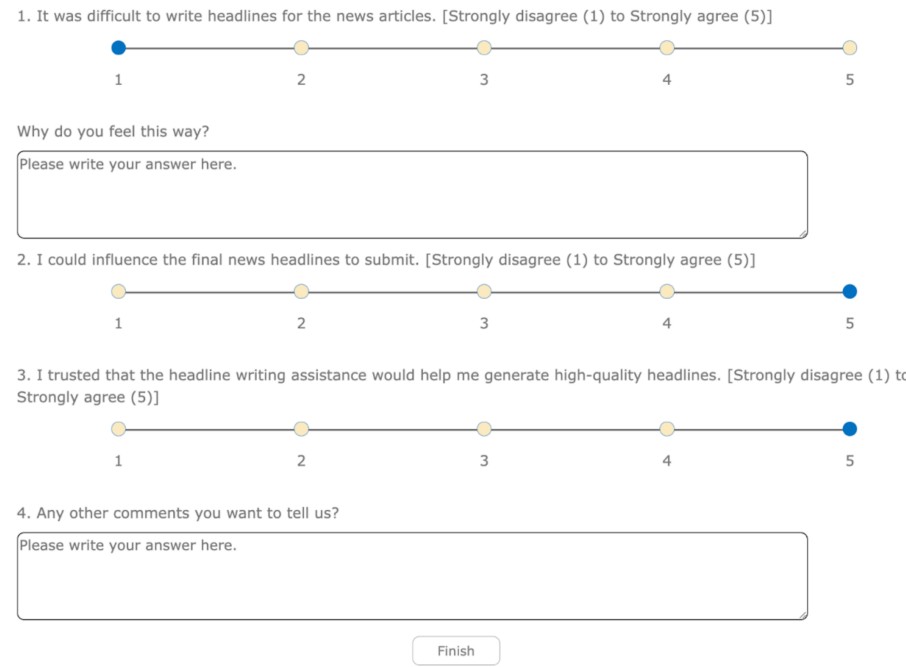

**The "Finish" button is clickable after you finish all the questions.**

1. It was difficult to write headlines for the news articles. [Strongly disagree (1) to Strongly agree (5)]

1        2        3        4        5

Why do you feel this way?

Please write your answer here.

2. I could influence the final news headlines to submit. [Strongly disagree (1) to Strongly agree (5)]

1        2        3        4        5

3. I trusted that the headline writing assistance would help me generate high-quality headlines. [Strongly disagree (1) to Strongly agree (5)]

1        2        3        4        5

4. Any other comments you want to tell us?

Please write your answer here.

Finish

Figure G8: Interface of the post-study survey.