# OpenReview forum: "Harnessing the power of LLMs: Evaluating human-AI text co-creation through the lens of news headline generation"
_EMNLP/2023/Conference — EMNLP 2023 Findings_

### Official Review · Reviewer_DQTz · 2023-07-27

**Soundness:** 3

**Excitement:**

3: Ambivalent: It has merits (e.g., it reports state-of-the-art results, the idea is nice), but there are key weaknesses (e.g., it describes incremental work), and it can significantly benefit from another round of revision. However, I won't object to accepting it if my co-reviewers champion it.

**Paper Topic And Main Contributions:**

This paper adopts three interactive types of guidance, selection, and post-editing to evaluate the human-AI text co-creation in the news headline generation. The work experiments and analyzes the combination of three common interaction types in human-AI interaction for headline generation, and provides design guidelines and human-evaluated datasets. I think the idea is valuable, but it lacks the sufficient experiments and in-depth analysis.

**Reasons To Accept:**

- The exploration of human-AI text co-creation is valuable and could be of interest to researchers in the community.
- The work provides a dataset of human-rated news headlines.

**Reasons To Reject:**

- The paper is lacking some deeper discussion on the experimental results of using their method.  E.g., why post-editing has no significant effect and why the manual condition has the lowest quality, which are counterintuitive. Since the manual condition's effect is the worst, then the quality of the condition combined with post-editing should be reduced with a high probability.
- I don't think 20 articles are enough to analyze the actual effects of different types, and instead of increasing the number of participants, consider increasing the number of different articles. As a straightforward example, choosing the best of the three outputs generated by the LLM, the results of 10 participants would not vary excessively.
- The experiment is not sufficient. I am curious whether the accuracy of perspectives selection will have a significant impact on the quality of generated titles. If there is a task difficulty label, whether the effects of different types under different difficulty levels are considered, and whether the number of different task difficulty levels is balanced.

**Reproducibility:**

3: Could reproduce the results with some difficulty. The settings of parameters are underspecified or subjectively determined; the training/evaluation data are not widely available.

**Reviewer Confidence:**

4: Quite sure. I tried to check the important points carefully. It's unlikely, though conceivable, that I missed something that should affect my ratings.

---

> ### Author Rebuttal · Authors · 2023-08-29
>
> Thank you, Reviewer DQTz, for your comprehensive review and constructive suggestions, particularly your recognition of the value in exploring human-AI text co-creation and the contribution of our human-rated news headlines dataset. Given the limitations of space and the emerging nature of this field of study, your feedback has been instrumental in identifying areas for improvement, and we have made a concerted effort to address each of your points below.
>
> Insufficient Experimental Discussion
>
> We appreciate the insightful comments concerning the depth of our experimental results section.
>
> Our results show that AI-generated headlines already possess a high level of quality, which enables them to outperform manually crafted headlines within a constrained time frame. Furthermore, the high initial quality of the AI-generated headlines likely explains why post-editing does not lead to significant improvements.
>
> To address the notion that "the quality of the condition combined with post-editing should be reduced with a high probability," we will delve into the differential experiences that manual crafting and post-editing offer. Specifically, post-editing permits humans to focus primarily on fine-tuning and minor adjustments, as the AI has already done much of the initial work. This stands in contrast to the more labor-intensive process of generating a headline manually from scratch.
>
> We will incorporate these insights for a more comprehensive interpretation of the findings in the Discussion.
>
> Limited Number and Types of Articles
>
> We appreciate the critique on the limited number of articles used for evaluating different conditions. The scope of our study was a first step beyond Cheng et al.'s Wizard-of-Oz approach, incorporating real system interactions. Given the constraints of our study's scale, our primary focus was understanding the differences between the three interactive variants rather than exhaustively exploring variations in articles and headline tasks. Future research should build on this  study through the inclusion of a more diverse set of articles, a longer-term assessment, and a varied participant pool. We will add these important points to the Limitations and Future Work. These combined efforts will offer a more nuanced understanding of the model's performance across a range of scenarios.
>
> Experimental Limitations and Task Difficulty
>
> Thank you for pointing out the issue of "accuracy of perspective selection" and the potential impact of task difficulty. We acknowledge the limitation of not having evaluated the model's accuracy for generating perspectives, despite employing a state-of-the-art system available at the time and undertaking prompt engineering coupled with qualitative analysis to enhance the quality of generated perspectives. We are open to sharing the user-selected perspectives in our dataset for further scrutiny. As a future improvement, we are considering the incorporation of user-specified keyword options, which would enable us to assess the performance of system-generated keywords.
>
> Regarding task difficulty, we endeavored to minimize variability by choosing articles of similar lengths and from common domains, and all participants worked with the same set of articles. Nevertheless, we acknowledge that task difficulty could be a relevant factor in future, more expansive studies.
>
> We hope that our responses successfully address Reviewer DQTz's concerns to clarify the scope and limitations of our research. Your feedback is greatly appreciated and will help strengthen our revised paper. Thank you for considering our work for inclusion in EMNLP 2023!

---

### Official Review · Reviewer_Nsef · 2023-08-04

**Typos Grammar Style And Presentation Improvements:** 1. "four" --> "three" L367
**Soundness:** 4

**Excitement:**

3: Ambivalent: It has merits (e.g., it reports state-of-the-art results, the idea is nice), but there are key weaknesses (e.g., it describes incremental work), and it can significantly benefit from another round of revision. However, I won't object to accepting it if my co-reviewers champion it.

**Paper Topic And Main Contributions:**

This paper proposes a study that compares how different human-AI interaction strategies (guidance, selection and post-editing) compare on the task of news headline generation based on three measures -- headline quality, task effort and perceived control and trust, based on prior work (Cheng et. al, 2022). The work operationalizes Cheng. et al (conducted on a simulated task) on a real-world text-generation setting. These interactions are conducted in a controlled setting with curated participants, where the participants involved in the interactions are separate from the evaluators.

**Questions For The Authors:**

QA. Novelty: Can you specify exact points of distinction from Cheng et. al, since the interactions paradigms and evaluation measures are both borrowed from that work?

QB. How often did you notice that GPT3 generated headlines were absurdly factual? Could it be that factuality or verbosity were factors that confounded the *quality* of headlines? How exactly would you define quality in this case?

**Reasons To Accept:**

RA1: Timeliness: The paper provides a timely analysis of different human-AI interaction paradigms for a co-writing task. Given the prevalence of GPT3 in writing tasks in the present, the paper is a good attempt to quantify the benefits on interacting with LLMs along three measure axes.

RA2: Study Design: The authors did a great job at designing a controlled experiment study, as well as presenting details of their study.

**Reasons To Reject:**

RR1: Evaluation Measures: Evaluators were ranking generations by GPT3.5, Human-only and three other assisted settings. However, there is no control for the reasons evaluators' rankings. GPT3.5 is known to be very verbose even in short generations, and for settings like news which GPT3.5 would have probably even been trained on, there is no guarantee that their generations were actually better or just verbose for that matter. While the paper reports these reasons in Table 4 (appendix), they are not accounted during the actual quality evaluation, which makes the actual evaluation underspecified.

RR2: Interaction strategies: The paper only uses the first three interaction strategies from Cheng. et al. The writers are not allowed to prompt or interact with the model, and is thus only constrained to guide the generation (Setting 2) based on the options they are provided (which are also generated via prompting). Firstly, not only is this constrained to the quality of keyword generations by GPT itself, there is no option for writers to abstain from GPT's selections (in case of low-quality generations). Furthermore, given that annotators are not actually allowed to interact with the model (Setting 4/5 in Cheng et. al), model guidance in terms of keywords is not a realistic guidance setting.

Update post rebuttal: The authors have addressed RR1 with satisfactory arguments.

**Reproducibility:**

3: Could reproduce the results with some difficulty. The settings of parameters are underspecified or subjectively determined; the training/evaluation data are not widely available.

**Reviewer Confidence:**

4: Quite sure. I tried to check the important points carefully. It's unlikely, though conceivable, that I missed something that should affect my ratings.

---

> ### Author Rebuttal · Authors · 2023-08-29
>
> We are grateful for the detailed feedback and constructive criticisms offered by Reviewer Nsef, especially the remarks on the timeliness of our work, our meticulous study design and detailed presentation. Despite the constraints of paper length and the emerging nature of this research area, your insights have been instrumental in shaping impactful revisions that will enhance both the quality and depth of our paper. Herein, we respond to each of the points.
>
> Evaluation Measures Underspecified, especially for Verbosity and Factuality
>
> Thank you for highlighting the need for rigorous evaluation measures in our study. To address this, we've employed the Taste-Attractiveness-Clarity-Truth (TACT) framework, a well-established and standardized rubric commonly used in journalism education (http://www.columbia.edu/itc/journalism/isaacs/client_edit/Headlines.html), as our measure for evaluating headline quality. Referenced in both the main text and Appendix G2, this framework specifically addresses critical quality metrics such as clarity and truthfulness. It prescribes guidelines across four dimensions to limit evaluator subjectivity (Section 3.2.2) and preemptively tackles potential pitfalls like verbosity and factual inaccuracies (with clarity and truthfulness) that may occur with GPT-3.5.
>
> We also leveraged the expertise and experience of our evaluators to enhance the validity of our assessments. While we believe the TACT framework provides a robust basis for evaluation, we acknowledge its limitations in capturing all nuances of a model's performance. Therefore, we will discuss these limitations and propose directions for refining the evaluation criteria in Future Work.
>
> Limited Interaction Strategies
>
> Thank you for your observation, which dovetails with the 'Guidance Variant is Highly Constrained' concern raised by Reviewer YiDy. We intentionally concentrated on the first three interaction strategies outlined in Cheng et al. to steer clear of the complexities inherent in multiple interaction rounds, which are generally better suited for longer text editing interactions. Nevertheless, we acknowledge the need for more adaptable interaction mechanisms and will actively consider incorporating such elements in our future research.
>
> Novelty Relative to Cheng et al.
>
> Our study builds upon Cheng et al.'s Wizard-of-Oz framework by employing an actual AI-powered system for text news headline generation and editing. This methodological advancement enables us to capture more realistic and reliable insights into human-AI co-creation, while still respecting the original framework. Building a functional AI system enables measurements that are more reflective of genuine human-AI co-creation with realistic AI performance. We will clarify this novelty in comparison to Cheng et al. in the paper.
>
> We hope that the above responses adequately address the queries and concerns put forth by Reviewer Nsef. Your feedback has been invaluable in clarifying both the strengths and areas for improvement in our paper. Thank you for your time and consideration in reviewing our submission for EMNLP 2023!

---

### Official Review · Reviewer_YiDy · 2023-08-05

**Soundness:** 3

**Excitement:**

3: Ambivalent: It has merits (e.g., it reports state-of-the-art results, the idea is nice), but there are key weaknesses (e.g., it describes incremental work), and it can significantly benefit from another round of revision. However, I won't object to accepting it if my co-reviewers champion it.

**Missing References:**

In the related works, might be relevant to add “Clickbait? Sensational Headline Generation with Auto-tuned Reinforcement Learning” (Xu et al., 2019)

**Paper Topic And Main Contributions:**

This paper studies human-AI text co-creation (i.e., humans writing with different levels of AI assistance) in the context of news headline generation through a well-designed human subject evaluation study. They consider what types of assistance are most effective for helping humans write headlines, how the types of assistance affect the perceived effort of writing, and how crowd workers’ interactions with LLMs affect their trust/feelings of ownership of the headlines.

**Questions For The Authors:**

A. Why do you choose that specific combination of selection, guidance + selection, etc.? Why not selection + post-editing or just each individual method by themselves?
B. Newsline generation/editing is subjective to the audience and the selections, edits, and prompting depend heavily on the person and may not be preferred by others. Can you discuss how this may impact your experiment/results?

**Reasons To Accept:**

This paper presented a well-designed study aside from a few points in an impactful domain, given the increasing popularity of LLMs in downstream tasks such as news headline generation. There is signficant discussion of results and many behaviors of participants (e.g., post-editing) is well-documented/explained.

**Reasons To Reject:**

Generally, the weaknesses of this paper can be summarized as:
1. The guidance variant is highly constrained, only allowing certain keywords to be emphasized. Ideally, this should be user-specified.
2. Subjectivity is not addressed in this work - indviduals and audiences have preferences as to what constitutes a good headline
3. The scale of the study made it difficult to observe significant trends, especially for RQ2 and RQ3. Furthermore, since experience varied significantly, many trends seemed obscured.

One additional note: I would have liked to see how different models (e.g., GPT-3, fine-tuned BART for headline generation) impacted perceived difficulty of writing as well as quality. This would provide insights into whether more expensive models are actually beneficial for this task.

**Reproducibility:**

N/A: Doesn't apply, since the paper does not include empirical results.

**Reviewer Confidence:**

4: Quite sure. I tried to check the important points carefully. It's unlikely, though conceivable, that I missed something that should affect my ratings.

---

> ### Author Rebuttal · Authors · 2023-08-29
>
> We appreciate the insightful comments and constructive suggestions offered by Reviewer YiDy, especially for recognizing that this study represents an initial exploration in a highly impactful domain, as well as for the points regarding the study design, discussion of results and documentation of participants’ behaviors. Although it was not feasible to provide a comprehensive treatment of all relevant aspects given the limitations of paper length and the nascent nature of this research area, deeply engaging with the comments has resulted in a number of impactful revisions we will make to the paper. Below, we address each raised point.
>
> Guidance Variant is Highly Constrained
>
> We appreciate this insightful observation. The purpose of the guidance variant is to offer a controlled scenario that both allows for guiding the headline generation and assists the user to do so by suggesting possible directions (keywords). This is designed to help users unfamiliar with how to interact with an LLM, especially considering the study's timeframe in July 2022, and give some suggestions of possible perspectives (based on LLM analysis of the article content) without the user needing to come up with these alone. However, we recognize the benefit of further supporting user-specified keywords, allowing a more flexible interaction. We will update the Discussion to include this suggestion of providing users the choice between system-generated and user-generated keywords as a future variant.
>
> Subjectivity Not Addressed, and Subjectivity Based on Audience
>
> We concur that subjectivity is a significant factor, particularly in the evaluation of headlines. To counteract this, we employed the Taste-Attractiveness-Clarity-Truth (TACT) framework as a standardized evaluation rubric. This framework offers concrete benchmarks across four dimensions to minimize individual evaluator subjectivity (Section 3.2.2). Despite having a robust evaluation framework and recruiting evaluators with editorial experience, this subjectivity may account for discrepancies between human-selected and solely AI-generated headlines. We will elaborate on this aspect in the Discussion.
>
> Scale of the Study
>
> We acknowledge the limitation in scope due to the specialized skills required for both writing and evaluating headlines. However, our sample size (40) is consistent with similar human-AI interaction studies, such as the 22 participants in the study conducted by Li et al. (2023). The aim was to provide an initial understanding of different interaction variants, and future studies should expand on this with larger participant samples and more diverse skill sets. We include this as a suggestion for future work and in the Limitations.
>
> Evaluating Different Models
>
> We acknowledge the importance of evaluating different language models, while our study's primary focus was on human-AI interaction. Building upon Cheng et al. work, which used a 'Wizard of Oz' setup, we employed a real-world system with GPT-3.5 with state-of-the-art performance at the time of our research. Adding performance as an additional factor (and comparing additional models) would have complicated the study design making it hard to understand our key factor: the three interaction variants. We agree that examining the impact of different models (performance) is a valuable avenue for future work and we will add this to the Limitations and Future Work sections.
>
> Choice of Combination Methods
>
> Our goal was to create a continuum of human-AI interaction methods ranging from less to more control. The layering approach of selection, guidance, and post-editing was informed by prior literature (Cheng et al.) and allows us to assess the impact of each new layer by comparing adjacent conditions. This will be further clarified in the paper.
>
> We hope these responses address the concerns of Reviewer YiDy and help clarify the contributions and limitations of our work. Thank you for considering our paper for EMNLP 2023!
>
> Reference
>
> Li, Daniel, et al. "Improving Automatic Summarization for Browsing Longform Spoken Dialog." Proceedings of the 2023 CHI Conference on Human Factors in Computing Systems. 2023.

---

### Meta-Review · Area_Chair_Devt · 2023-09-15

**Recommendation:** 3

**Metareview:**

The reviewers generally agreed that the studies were well designed and the discussion was well written. Reviewer YiDy appreciated both. Reviewer YiDy also emphasized the timeliness of this analysis of different Human-AI Interaction paradigms. Reviewer DQTz appreciated the human-rated news dataset.

Reviewer YiDy had a valid concern that the guidance variant is highly constrained and it should be user-specified. The authors acknowledged it and suggested making edits to the discussion to reflect that. They also had a concern that subjectivity remains unaddressed in the work.The authors pointed to their use of TACT framework. Finally they had a concern about the scale of the study, and the authors pointed out that the same size is consistent with the prior work.

Reviewer Nsef also had similar concerns about lack of control in evaluation metrics as well as not allowing users to prompt/interact with the model. The authors also made the novelty relative to Cheng et al.’s wizard-of-oz framework, and pointed out that they used an actual AI-powered system for news text generation.

Reviewer DQTz had most notable concerns about the methodology including limited number of articles and task difficulty balance. The authors acknowledge it in the rebuttal.

---

### Decision · Program_Chairs · 2023-10-07

**Decision:**

Accept-Findings

**Comment:**

The reviewers generally agreed that the studies were well designed and the discussion was well written. Reviewer YiDy appreciated both. Reviewer YiDy also emphasized the timeliness of this analysis of different Human-AI Interaction paradigms. Reviewer DQTz appreciated the human-rated news dataset.

Reviewer YiDy had a valid concern that the guidance variant is highly constrained and it should be user-specified. The authors acknowledged it and suggested making edits to the discussion to reflect that. They also had a concern that subjectivity remains unaddressed in the work.The authors pointed to their use of TACT framework. Finally they had a concern about the scale of the study, and the authors pointed out that the same size is consistent with the prior work.

Reviewer Nsef also had similar concerns about lack of control in evaluation metrics as well as not allowing users to prompt/interact with the model. The authors also made the novelty relative to Cheng et al.’s wizard-of-oz framework, and pointed out that they used an actual AI-powered system for news text generation.

Reviewer DQTz had most notable concerns about the methodology including limited number of articles and task difficulty balance. The authors acknowledge it in the rebuttal.